# G9a in Cancer: Mechanisms, Therapeutic Advancements, and Clinical Implications

**DOI:** 10.3390/cancers16122175

**Published:** 2024-06-08

**Authors:** Yuchao Ni, Mingchen Shi, Liangliang Liu, Dong Lin, Hao Zeng, Christopher Ong, Yuzhuo Wang

**Affiliations:** 1Department of Urology, West China Hospital, Sichuan University, Chengdu 610041, China; yni@prostatecentre.com; 2Vancouver Prostate Centre, Vancouver, BC V6H 3Z6, Canada; ashi@prostatecentre.com (M.S.); lliu@prostatecentre.com (L.L.); dlin@prostatecentre.com (D.L.); ywang@bccrc.ca (Y.W.); 3Department of Urologic Sciences, University of British Columbia, Vancouver, BC V5Z 1M9, Canada; 4Department of Experimental Therapeutics, BC Cancer, Vancouver, BC V5Z 1L3, Canada

**Keywords:** G9a, EHMT2, histone methyltransferase, prostate cancer, epigenetic inhibitor

## Abstract

**Simple Summary:**

In this review, we have summarized the crucial role of G9a in cellular biology and disease. By synthesizing recent research discoveries, we delve into the functions of G9a in gene expression regulation, cell proliferation, and biological development. Additionally, we explore its potential involvement in cancer and discuss its prospects as a therapeutic target specifically in prostate cancer. In summary, this review provides a comprehensive perspective on G9a, enhances understanding of the significance of G9a in biology and cancer, and offers insights into future research and drug development.

**Abstract:**

G9a, also named EHMT2, is a histone 3 lysine 9 (H3K9) methyltransferase responsible for catalyzing H3K9 mono- and dimethylation (H3K9me1 and H3K9me2). G9a contributes to various aspects of embryonic development and tissue differentiation through epigenetic regulation. Furthermore, the aberrant expression of G9a is frequently observed in various tumors, particularly in prostate cancer, where it contributes to cancer pathogenesis and progression. This review highlights the critical role of G9a in multiple cancer-related processes, such as epigenetic dysregulation, tumor suppressor gene silencing, cancer lineage plasticity, hypoxia adaption, and cancer progression. Despite the increased research on G9a in prostate cancer, there are still significant gaps, particularly in understanding its interactions within the tumor microenvironment and its broader epigenetic effects. Furthermore, this review discusses the recent advancements in G9a inhibitors, including the development of dual-target inhibitors that target G9a along with other epigenetic factors such as EZH2 and HDAC. It aims to bring together the existing knowledge, identify gaps in the current research, and suggest future directions for research and treatment strategies.

## 1. Introduction

Epigenetic modifications represent the regulation of gene expression without altering the DNA sequence. They contribute to various biological processes and disease states, including cancer [1]. Aberrant epigenetic reprogramming associated with profound transcriptional changes has been consistently identified across cancer types, correlated with tumor progression and poor prognosis [2,3,4]. Among the known epigenetic alterations, histone post-translational modifications emerge as crucial players in orchestrating chromatin status and its corresponding transcriptional networks [5].

In particular, histone methylation has gained extensive attention as the subject of a thorough investigation. Its significance has been revealed in multiple biological and molecular processes, including cell cycle regulation, gene expression regulation, and tumorigenesis [6,7]. Histone methylation is precisely balanced by the function of histone methyltransferases (HMTs, writers) and demethylases (erasers) [8]. Notably, among these regulatory players, G9a, an HMT catalyzing mono- and dimethylation at histone 3 lysine 9 (H3K9me1 and H3K9me2), has been evidenced to hold a pivotal role in carcinogenesis and aggressiveness across cancer types [9,10].

Despite recent reviews on G9a primarily focusing on its structure and function, as well as its relationship with stemness, drug resistance, and roles in pediatric and adult brain tumors, significant progress has also been made in studying G9a in the context of prostate cancer and the tumor microenvironment [11,12,13]. Herein, in this review, we offer comprehensive insights into the roles of G9a in biology and the current research landscape that highlights the involvement of G9a in tumorigenesis, progression, and cancer stemness in different types of cancer. We place particular emphasis on the recent findings in the field of prostate cancer. Furthermore, we aim to conclude the information on G9a inhibitors, pinpoint the existing gaps in the prostate cancer research field, and shed light on future research directions.

## 2. Molecular Structure and Function of G9a in Epigenetic Modification

G9a, alternatively known as KTM1C, EHMT2, or BAT8, was first identified by Milner in 1993 [14]. As a nuclear lysine methyltransferase, G9a belongs to the SET HMT family. It exists in the form of a homodimer or a heterodimer with another HMT G9a-like protein (GLP) [15]. G9a, together with GLP, forms a dimer complex that is crucial for catalyzing histone markers H3K9me1 and H3K9me2. Notably, overexpression of G9a, and not of its related protein GLP, has often been associated with a more aggressive phenotype in cancer [16]. H3K9 methylation is dynamically regulated by various enzymes and is involved in multiple biological processes, including development, differentiation, and response to environmental signals. This modification plays a critical role in regulating gene expression by contributing to the formation of heterochromatin and the repression of specific genomic regions [10]. Since H3K9 methylation primarily contributes to gene silencing, G9a was generally regarded as an epigenetic repressor [17]. However, certain studies have reported that G9a could generate the trimethylation of H3K9 (H3K9me3) in long-time incubation in vitro [18,19]. Given that the blocking effect of the hydroxyl group at Y1067 in the SET domain of G9a hinders the alignment of the dimethylamine in H3K9me2 and prevents the formation of H3K9me3, G9a is considered as the major H3K9me1 and H3K9me2 methyltransferase [20,21].

Structurally, G9a comprises several domains critical for its function: an activation domain, a glutamate-rich region, nuclear localization signals, a cysteine-rich region, seven ankyrin repeats, and a SET domain responsible for its methyltransferase activity from the N-terminus to the C-terminus (Figure 1). The SET domain, named after Su(Var)3-9, Enhancer-of-zeste, and Trithorax, is a common shared structural feature among all six members of the H3K9 methyltransferase family [10]. The SET domain contains an evolutionarily conserved sequence of 130 to 140 amino acids and contributes to the catalytic activity of G9a [22]. Wu et al. reported the indispensability of the tyrosine residue Y1067 for the catalytic function of G9a as an HMT [21]. Similar to that of other H3K9 methyltransferases, the SET domain within G9a also encompasses pre-SET and post-SET domains [10]. The pre-SET domain functions as a structural stabilizer that establishes multiple interactions with the surfaces of core-SET domains; the post-SET domain facilitates the formation of hydrophobic channels.

A total of 6 contiguous ankyrin repeats, each consisting of 33 amino acids, form a module in G9a that is capable of binding mono- and dimethyl lysine, thereby mediating protein–protein interactions [23,24]. The specificity of the ankyrin repeat domain of G9a is comparable to that observed in other H3K9 methylation binding protein modules, including the chromodomain (of SUV3H1/H2) and the tudor domain (of SETDB1, MET-2), which could specifically bind lysines methylated at histone tail residues [25,26,27]. This domain could operate as a scaffold, facilitating the recruitment of other corepressors through the recognition of H3K9 methylation markers [23].

## 3. Embryonic Development and Tissue Differentiation

Embryonic development and tissue differentiation are intricately regulated by epigenetic mechanisms, among which the histone methyltransferase G9a plays a pivotal role. The enzymatic activity of G9a, particularly its methylation of histone H3 on lysine 9 (H3K9me1 and H3K9me2), is crucial for the transcriptional repression of genes involved in developmental processes across various organisms [28,29]. Studies in mice have demonstrated that depletion of G9a drastically reduced H3K9 methylation in embryonic stem (ES) cells. Consequently, these G9a-deficient ES cells displayed severe growth inhibition and early lethality [28]. Surprisingly, G9a depletion in ES cells led to a significant increase in methylated H3 lysine 4 (H3K4), a mark associated with gene activation, regardless of H3K9, suggesting a complex regulatory network balancing gene expression during development [29].

Further research has elucidated the pivotal role of G9a in bone formation and immune cell differentiation. For example, the interaction of G9a with RUNX2 is essential for normal osteoblast development; its regulatory influence on the Twist gene is vital for osteogenesis [30,31]. Specifically, Hisashi et al. reported that G9a could endogenously bind to RUNX2-related regions; depletion of G9a in osteoblasts hindered the binding of RUNX2 to the osteocalcin promoters, resulting in the abnormal formation of cranial bones [30]. In addition, Higashihori et al. demonstrated that G9a regulated osteogenesis via Twist gene repression. They found that Twist1 and Twist2 regulatory regions contained substantial H3K9me2 catalyzed by G9a in primary osteogenic mesenchyme from calvaria. Moreover, pharmacological inhibition of G9a decreased several osteogenic markers and prevented skeletal differentiation in vivo [31].

The significance of G9a in embryonic development parallels its role in cancer biology [16]. In cancer, aberrant G9a activity can lead to improper gene silencing, similar to its role in silencing developmental genes. However, in the context of cancer, this results in the repression of tumor suppressor genes and the activation of pathways that promote cancer cell proliferation, survival, and metastasis. Understanding the function of G9a in developmental biology, particularly its ability to dictate cell fate and maintain cellular identity through epigenetic modifications, informs its potential to drive oncogenesis when dysregulated.

G9a could also contribute to the T helper (Th) cell differentiation. Depletion of G9a in Th precursor cells was observed to impair their ability to develop into Th2 cells under exposure to chronic infection. G9a depletion might elevate the IL-17A expression via the loss of H3K9me2 in the IL-17A locus, thereby mitigating the lineage plasticity capacity of Th precursor cells [32].

Understanding the dual roles of G9a in promoting normal development and its aberrant expressions contributing to cancer allows researchers and clinicians to gain a better appreciation of the potential for targeting G9a in cancer therapy.

## 4. Pivotal Role of G9a in Cancers

Beyond its fundamental role in embryonic development, G9a has been recognized as an upregulated player in various cancers, including breast, lung, and liver tumors. There is a growing consensus on the significant impact of G9a-induced aberrant epigenetic reprogramming on tumorigenesis, disease progression, autophagy, apoptosis, lineage plasticity, and hypoxia adaptation (Figure 2). Understanding how G9a contributes to cancer development and progression is critical for advancing targeted epigenetic therapies aimed at enhancing patient outcomes.

### 4.1. Dysregulation of Tumor Suppressor Genes

The inactivation of tumor suppressor genes is a pivotal step in cancer development. G9a-mediated H3K9 methylation plays a central role in the repression of tumor suppressor genes, promoting carcinogenesis. G9a overexpression leads to increased H3K9 methylation, which in turn attracts proteins such as heterochromatin protein 1 (HP-1), promoting the formation of compact, transcriptionally inactive chromatin structures [10,33]. This silencing mechanism contributes to the downregulations of tumor suppressor genes across various cancer types, facilitating tumor growth and progression. Notably, Francesco et al. demonstrated that G9a under hypoxic conditions in breast cancer could downregulate key genes, including ARNTL and CEACAM7, highlighting common pathways cancer cells utilize to evade growth control [34]. Similarly, in liver cancer, overexpression of G9a is linked to the silencing of RARRES3, promoting tumor aggressiveness via the miR-1/G9a/RARRES3 signaling axis [35]. Additionally, a study conducted by Lee et al. showed that G9a significantly downregulated the expression of the tumor suppressor gene RUNX3 in gastric cancers [36]. They found that upregulated G9a significantly inhibited the nuclear localization of RUNX3 and attenuated the RUNX3 expression during tumor progression. Similarly, overexpressed G9a is found in cholangiocarcinoma (CCA) as a silencer to the tumor suppressor gene, LATS2, mediating tumor development [37]. Ma et al. reported that inhibition of G9a in CCA significantly reduced the level of H3K9me2 and restored the expression of LATS2, leading to YAP inhibition.

Moreover, the capacity of G9a extends beyond histone methylation to directly methylate non-histone proteins, such as p53, further illustrating its broad role in cancers. Huang et al. found that G9a specifically dimethylated p53 at Lys373, resulting in the inactivation of p53 [38]. These data suggested that G9a has the potential regulations directly on genes themselves, regardless of its catalytic function of histone methylation. In subsequent studies, additional targets have been identified to serve as direct substrates for G9a methylation, including transcription factors and related transcription co-regulatory factors, such as p53, ERα, CEBP, HDAC1, and HIF1α, further illustrating the extensive influence of G9a in cancer biology [38,39,40,41,42].

### 4.2. Promote Tumor Proliferation and Progression

Elevated G9a expression is strongly associated with an aggressive cancer phenotype, significantly influencing disease progression and metastasis through its epigenetic regulatory functions. G9a exerts a profound impact on cancer biology by modulating key pathways crucial for tumor growth and dissemination, such as epithelial–mesenchymal transition (EMT), mTOR, and Wnt/β-catenin signaling.

In lung cancer, numerous studies underscore the pivotal role of G9a in promoting aggressiveness through its impact on the EMT pathway [43,44,45,46,47,48]. Chen et al. demonstrated that G9a knockdown not only ablated H3K9me2 expression but also suppressed the EMT-related gene Ep-CAM by decreasing the recruitment of the transcriptional cofactors HP1, DNMT1, and HDAC1 to the promoter region [44]. Similarly, Hu et al. found that G9a and HDAC jointly regulated Snail-induced E-cadherin suppression to enhance migration and invasion in lung and liver cancers [47,49]. In breast cancer, the involvement of G9a in the EMT pathway has been reported to contribute to heightened cancer growth and metastatic potential. For example, Wozniak et al. highlighted the facilitative role of G9a in enhancing the EMT pathway through silencing desmocollin 3 (DSC3) and MASPIN via H3K9 methylation [50]. In addition, G9a could regulate Snail during EMT in breast cancer, leading to the suppression of E-cadherin expression through promoter methylation. Dong et al. reported that in vivo and in vitro knocking out of G9a in breast cancer could rescue E-cadherin expression and concurrently inhibit cancer proliferation and progression [51]. Additionally, this G9a–Snail–E-cadherin axis was also reported in the head and neck squamous cell carcinoma (HNSCC). Liu et al. demonstrated that the G9a protein could be essential for the induction of EMT in HNSCC through the G9a–Snail–E-cadherin axis to promote tumor cell migration and tumorphere formation [52].

Beyond EMT, several other pivotal pathways have been investigated to be regulated by G9a to promote tumor cell proliferation in various cancers [44,48,53]. For instance, Zhang et al. observed overexpression of G9a in 43.2% of non-small cell lung cancer (NSCLC) tissues. They found that G9a suppression could dramatically repress NSCLC tumor growth through Wnt signaling pathway inhibition [44]. Also, Sun et al. reported that elevated G9a expression was correlated with focal adhesion kinase activation via NF-κB signaling [48]. Another study exploring the role of G9a in NSCLC uncovered its contribution to tumor cell growth and invasion through the repression of CASP1 [46]. In gastric cancer, Yin et al. found that knockdown of G9a or treatment with G9a inhibitor BIX01294 significantly reduced cell growth by inducing cell cycle arrest and autophagy. Mechanically, mTOR expression was linked to promoter methylation and enrichment for H3K9me1 and H3K9me2. G9a knockdown decreased the H3K9 methylation level at the mTOR promoter in gastric cancer cells, leading to inhibited tumor growth. Notably, they demonstrated an apparent decrease only in the H3K9me1 but not in the H3K9me2 level after G9a knockdown [53].

However, there remain other vital mechanisms associated with G9a in the process of tumor proliferation and progression. In gastric cancer, Kim et al. reported that G9a inhibition might lead to autophagic cell death under the endoplasmic reticulum (ER) stress via the HDAC–G9a and STAT3–G9a axes [54,55]. Simultaneously, Zhang et al. uncovered the association between G9a and the p21-Bim signal in breast cancer. They found that G9a inhibition by GA001, a potent G9a antagonist, significantly inhibited the MCF7 cell proliferation and survival by inducing autophagy and apoptosis dependent on the activation of AMPK [56]. Furthermore, Nakatsuka et al. demonstrated that G9a allowed DNA-damaged hepatocytes to escape apoptosis by suppressing Bcl-G [57]. They proposed the mechanism that DNA damage stimuli recruited G9a to the p53-responsive element of the Bcl-G gene. G9a enriched in the p53-responsive region significantly decreased the expression of the Bcl-G gene; therefore, it facilitated the evasion of p53-induced apoptosis, which might contribute to the tumor initiation and progression.

### 4.3. Cancer Lineage Plasticity

Accumulating evidence indicated that cancer cells have the ability to switch their cellular phenotype through lineage plasticity, transitioning themselves into a more undifferentiated, pluripotent, embryo-like, stem-like state. This adaptive mechanism facilitates tumor proliferation, drug resistance, and metastasis but remains unaddressed by conventional therapeutic measures in clinical practice. In concordance with its role in embryonic development, G9a is expected to be crucial to lineage plasticity within the field of cancer. Several studies have indeed substantiated this hypothesis, demonstrating that G9a could maintain the self-renewal ability of cancer cells and manifest cancer lineage plasticity as a stem-like phenomenon [52,58,59,60,61,62,63,64]. For instance, in NSCLC, Pangeni et al. reported that knocking down of G9a upregulated a series of genes (CDYL2, DPP4, SP5, FOXP1, STAMBPL1, and ROBO1) through hypomethylated H3K9, consequently sustaining tumor-initiating cells and facilitating cancer metastasis [58]. They found that G9a inhibition might downregulate the expression of cancer stem cell markers and defected sphere-forming capacity, in vitro proliferation, and in vivo growth. Notably, the chromatin immunoprecipitation (ChIP) assay showed that G9a interacted with its target genes through H3K9me2, emphasizing the role of G9a in a novel therapeutic approach. Numerous studies have delved into the role of G9a and cancer lineage plasticity across various tumors, exploring how G9a regulates tumor stemness (e.g., CD133 expression). These investigations have revealed that G9a may play distinct roles in different types of tumors. Recent reviews have provided detailed overviews and summaries of these findings, highlighting the diverse functions of G9a in cancer stemness [11,12].

The dual role of G9a in cancer introduces a novel perspective on the intricacies of lineage plasticity. While it sometimes serves as a suppressor of cancer lineage plasticity, G9a concurrently promotes cancer stem cell-like features in certain conditions. Indeed, this duality not only highlights the diverse functions of G9a but also provides valuable insights for further exploration into its underlying mechanisms across different cancer types. A profound understanding of the functions of G9a holds the potential to uncover additional treatment strategies in cancer therapy in the future.

### 4.4. Hypoxia Adaption

Hypoxia is a common phenomenon in solid tumors, attributed to excessive oxygen consumption due to the aggressively proliferating, overactive metabolism of tumors and abnormal formation of blood vessels supplying the tumor [65]. Cells perceive the change of oxygen concentration through a family of transcription factors known as hypoxia-inducible factors (HIF-1) [66,67]. A typical cellular response to hypoxic conditions is the maintenance of oxygen homeostasis. Cancer cells initiate hypoxic molecular mechanisms, including activation of survival mechanisms, metabolic reprogramming, and angiogenesis during the process of hypoxia adaption [68]. Under hypoxia, G9a is demonstrated to be upregulated to suppress a series of genes through H3K9 methylation to promote cancer survival [69].

In breast cancer, Casciello et al. found that G9a was a key player in mediating the hypoxia-related response. Under hypoxic conditions, an increased level of G9a inhibited several genes, including ARNTL, CEACAM7, and more to promote cancer cell survival and tumorigenesis [34]. They further reported that G9a could repress CDH10 expression through histone methylation, promoting tumor progression under hypoxia [65]. In gastric cancer, Kim further elucidated that G9a might promote cell survival under hypoxia via the STAT3–G9a axis. Additionally, they employed an herbal formulation, SH003, to induce ER stress, inhibiting G9a activity and ultimately inducing autophagic cell death [55].

## 5. Pharmacological Targeting of G9a

Given that G9a plays a pivotal role as an HMT in multiple tumorigenesis processes, including dysregulation of tumor suppressor genes, inactivation of transcription factors, and adaption to hypoxia tumor microenvironment (TME), it has been extensively investigated as a promising therapeutic target in the past decades. Here, we summarize the current progress in the development of G9a inhibitors.

To date, G9a inhibitors have broadly fallen into two groups [70,71]. The first category comprises substrate (H3 peptide) competitive inhibitors, including BIX-01294 [72], UNC0224 [73], UNC0321 [74], UNC0638 [75], UNC0642 [76], EML741 [77], and A-366 [78] (Table 1). The second category encompasses the S-5′-Adenosyl-L-Methionine (SAM) cofactor competitive inhibitors, such as BIX-01338 [72], BRD9539 [79], and BRD4770 [79]. Unfortunately, none of these inhibitors have progressed to clinical trials as of now. Considering the superior selectivity and indicative lower toxicity of substrate competitive inhibitors for G9a, recent research efforts have predominantly focused on the development of G9a inhibitors based on the BIX01294 scaffold (quinazoline-based) [80].

BIX-01294, initially reported in 2007, is recognized as the earliest small-molecule inhibitor specifically targeting G9a HMT activity and significantly reducing H3K9me2 expression [72]. As reported by Kubicek et al., BIX-01294, which is structured as a diazepin-quinazolin-amine derivative, emerged as a highly selective compound selected from a pool of over 125,000 preselected candidates and demonstrated notable efficacy in inhibiting G9a catalytic function and the generation of H3K9me2 in several cell lines. Numerous studies subsequently affirmed its effectiveness in inhibiting tumor growth by decreasing H3K9me2 expression levels [81,82,83]. However, the application of BIX-01294 is greatly limited by the narrow gap between the effective concentration for inhibiting H3K9me2 and the non-specific cytotoxic concentration [72,75]. Consequently, prompted by practical needs, UNC0638 was derived from the BIX-01294 structure through optimization. UNC0638 met expectations and has been reported to effectively reduce tumor growth in in vitro experiments across various cancer types [84,85,86,87].

However, subsequent research revealed poor pharmacokinetics for UNC0638 in in vivo experiments [75]. To overcome this shortage, Liu and his colleagues further modified UNC0637 and synthesized UNC0642 [76,88] (Figure 3). UNC0642 not only maintains high in vitro and cellular potency with low cell toxicity, like UNC0638, but also exhibits improved in vivo pharmacokinetic properties, rendering it suitable for animal studies [89,90]. Notably, A-366, distinguished by its indole core-based structure in contrast to UNC0642, a quinazoline core-based inhibitor, has progressed into the preclinical stage for leukemia [91]. This advancement is attributed to its heightened about ten-fold selectivity for G9a over GLP [78]. Moreover, several new G9a inhibitors have been developed in recent years, building on existing chemical structure cores or featuring entirely new structures, such as ZZM-1220 and SDS-347 [92,93].

Initially, G9a was found to participate collaboratively with DNMT1 in gene silencing through epigenetic regulation. In 2016, José-Enériz et al. employed a novel, potent, selective, and reversible drug, CM-272, which significantly suppressed the activity of G9a and DNMT1 based on UNC0642 [94]. Comprehensive functional validation of CM-272 was conducted, encompassing both in vitro and in vivo settings, revealing its high selectivity for G9a and DNMT1. The study showed that CM-272 effectively inhibited the expression of H3K9me2 and 5mC, subsequently inducing IFN-responsive tumor apoptosis. Moreover, CM-272 has undergone extensive validation for its noteworthy cytotoxicity across diverse tumor types, including cholangiocarcinoma, hematologic malignancies, bladder cancer, and prostate cancer (PCa) [94,95,96,97,98].

**Table 1 cancers-16-02175-t001:** Substrate (H3 peptide) competitive inhibitors of G9a reported in the literature.

Compound	Core	Application In Vivo	IC_50_s
BIX-01294 [72]	Quinazoline	No	G9a = 1.7 μM; GLP = 0.9 μM
UNC0224 [73]	Quinazoline	No	G9a = 15 nM; GLP = 20–58 nM
UNC0321 [74]	Quinazoline	No	G9a = 6–9 nM; GLP = 15–23 nM
UNC0638 [75]	Quinazoline	No	G9a < 15 nM; GLP = 19 nM
UNC0642 [76]	Quinazoline	Yes [76]	G9a ≈ GLP < 2.5nM
EML741 [77]	Quinazoline	No	G9a ≈ GLP: 23 nM; DNMT1 = 3.1 μM
A-366 [78]	Indole	Yes [91]	G9a = 3.3 nM; GLP = 38 nM
ZZM-1220 [92]	Quinazoline	No	G9a = 458 nM; GLP = 924 nM

Furthermore, Velez et al. recently developed a first-in-class G9a/GLP proteolysis targeting chimera (PROTAC) degrader 10 (MS8709) [99] and showed a promising therapeutic effect in vitro by inducing G9a and GLP degradation at 3 μM in several cell lines.

## 6. The Significance of G9a in Prostate Cancer

### 6.1. The Role of G9a as an HMT in Prostate Cancer

The overexpression of G9a was shown to be correlated with poor survival outcomes in castration-resistant prostate cancer (CRPC) patients [100]. Besschetnova et al. demonstrated that the genetic alteration and expression of G9a were elevated in clinical CRPC samples [100]. Moreover, they found that the downregulation of G9a transcriptionally deactivated oncogenic programs, resulting in a significant decrease in cell proliferation and metastasis in LNCaP, PC3, and xenograft models. He subsequently pharmacologically inhibited the enzymatic activity of G9a using UNC0642 and reverified this phenomenon. In previous studies, G9a was found to be able to methylate lysine 114 of LSD1, an H3K4 demethylase, leading to the activation of gene transcription in PCa cells. This activation was associated with the recognition and interaction of LSD1 with the chromatin-remodeling protein CHD1 [101]. This finding suggested that G9a might potentially establish a network with other epigenetic regulators in PCa, collectively influencing transcription through histone modification. AR inhibitors and androgen deprivation therapy (ADT) are crucial for the clinical treatment of prostate cancer. However, there is currently a lack of solid evidence indicating an interaction between G9a and AR, or even AR-V7. Although one study showed that H3K9 methylation can promote resistance to AR inhibitors in prostate cancer cells and that de-methylation may potentially restore drug sensitivity, it remains unclear whether there is a direct interaction between G9a and AR [102,103].

### 6.2. G9a–PRC2 Complex

Polycomb repressive complex 2 (PRC2) consists of core subunits, including Enhancer of Zeste homolog 2 (EZH2), suppressor of Zeste 12 (SUZ12), and embryonic ectoderm development (EED). This complex catalyzes mono-, di-, and trimethylation of histone 3 lysine 27 (H3K27me1, H3K27me2, and H3K27me3), extensively involved in PCa disease progression and drug resistance. As the catalytic component of the PRC2 complex, EZH2 has drawn widespread attention. Multiple studies have suggested that knocking down EZH2 expression in prostate cancer effectively lowers the levels of H3K27me3, thereby hindering the growth and progression of PCa [104,105,106,107]. Consequently, the development of EZH2 inhibitors has become a subject of significant interest among researchers, yielding promising outcomes.

Moreover, with further investigation into the structure of the PRC2 complex, G9a was found to be able to bind with the PRC2 complex, forming a novel G9a–PALI1–PRC2 structure and conducting dual H3K9 and H3K27 methylation epigenetic modifications [108,109]. This suggests that the combined therapy of G9a and EZH2 may yield more significant clinical efficacy in prostate cancer. They separately used two inhibitors targeting G9a (UNC0642) and EZH2 (EPZ-6438) for in vitro and in vivo experiments in LNCaP and C4-2B, revealing a significant synergetic effect. Indeed, there remains a significant gap in understanding the mechanism of G9a–PRC2 in PCa, underscoring the need for further research in this domain.

### 6.3. G9a/GLP-DNMT1 Complex

The tumor suppressor gene SPOP has been identified with a frequent mutation rate of 6–15% in prostate cancer, and it is associated with ERG rearrangements, aberrant androgen receptor (AR) activity, and globally elevated DNA methylation [110,111,112]. Zhang reported that in SPOP-mutant PCa cells, the aberrant upregulation of global DNA hypermethylation resulted from the inhibition of the SPOP-mediated polyubiquitination and degradation of the G9A/GLP-DNMT1 module [113]. This finding uncovered the mechanism by which G9a regulates the epigenetic landscape in PCa through DNA methylation. Moreover, it prompted the exploration of the efficacy of dual-targeted therapy against G9a and DNMT1 in PCa. CM-272, a dual inhibitor targeting G9a and DNMT1, has been reported by Filipa to effectively inhibit the expression of DNMT1, G9a, and H3K9me2 in prostate cancer. Consequently, CM-272 effectively suppressed tumor growth in PCa cell lines DU145, PC3, and 3D PCa models, demonstrating the promising anti-tumor effects in PCa.

## 7. Future Directions of G9a in Prostate Cancer

### 7.1. Immunotherapy

The integration of epigenetic inhibitors with immune checkpoint inhibitors (ICIs) has garnered attention in recent years, with an increasing number of studies delving into the underlying mechanisms by which epigenetic regulation modulates the tumor immune microenvironment: (1) Epigenetic modification has the potential to dysregulate the normal antigen presentation of tumor cells to induce immune evasion. Research suggests that epigenetic inhibitors can directly impact tumor cells by enhancing intracellular immune signaling pathways [114,115]. This includes elevating the expression of cell surface MHC class I molecules and restoring the cytotoxic effects of CD8+ T cells within the TME. (2) Epigenetic modification could influence the immunogenicity of tumor cells by altering the endogenous retrovirus (ERV) associated with sustained inflammatory stimulation. Studies have demonstrated that DNMT inhibitors and HDAC inhibitors can upregulate the expression of interferon pathways related to ERV by re-modulating the epigenetic landscape [97,116,117]. This induction creates a state in tumor cells similar to sustained viral stimulation, continuously triggering immune cell activation and thereby enhancing the efficacy of immunotherapy. (3) Beyond impacting tumor cells, epigenetic modification also regulates immune cells within the TME [118,119,120]. DNA methylation or histone modification plays an indispensable role not only in the formation and differentiation of immune cells but also in the induction of T cell exhaustion. This occurs through the reduction of T cell effector molecules and the overexpression of co-inhibitory factors such as PD-1, ultimately promoting tumor progression and drug resistance.

The rapid advancement of epigenetic inhibitors in modulating TME has prompted great efforts in combining these inhibitors with ICIs, revealing promising clinical efficacy in certain tumor types [114,121]. For instance, in melanoma, Kato et al. discovered that G9a played a crucial role in controlling tumor survival and progression by activating the WNT/β-catenin signaling pathway through stimulation of the microphthalmia-associated transcription factor (MITF). They observed that G9a contributed to the establishment of a ‘cold’ TME by inhibiting DKK1, an antagonist of the canonical WNT pathway. Furthermore, the effectiveness of G9a inhibitors in transforming a ‘cold’ immune microenvironment into a ‘hot’ one has been confirmed in an in vivo syngeneic mouse melanoma model, highlighting the clinic efficacy of combination therapy involving G9a inhibitors and immunotherapy [122]. In bladder cancer, Segovia et al. have confirmed the correlation between high G9a expression and poor survival in bladder cancer patients [97]. He verified that targeting G9a and DNMT methyltransferase activity with a dual inhibitor, CM-272, can induce cancer cell apoptosis and immunogenic cell death. Moreover, he inferred that epigenetic inhibitors have the capacity to convert cold immune tumors into hot immune tumors by modulating immune cells in the tumor microenvironment. This observation aligned with clinical findings that elevated G9a expression was correlated with poor efficacy of programmed cell death protein 1 inhibition in a cohort of patients with bladder cancer. These findings supported the potential opportunities of integrating immunotherapy with epigenetic inhibitors in the treatment of bladder cancer.

While G9a, as a critical HMT in prostate cancer, has gained attention only in recent years, it is reasonable to predict that G9a plays a crucial role in immune microenvironmental regulation, referring to the observations in other cancers. Unfortunately, there is currently limited research on the involvement of G9a in ‘cold’ TME of prostate cancer. Future studies, such as whether focusing on G9a in mediating immune escape, exploring its role in inflammatory signals, or elucidating its mechanisms in promoting immune cell exhaustion, require further validation for their effectiveness in prostate cancer. This will establish a solid foundation for the future combined regimen of G9a inhibitors and ICI in prostate cancer treatment.

### 7.2. Dual Inhibitor of G9a and EZH2

Inspired by the aforementioned study, G9a has been confirmed as a substrate of the PRC2 complex in prostate cancer, interacting with EZH2 to carry out epigenetic modification via the methylation of H3K9 and H3K27. The successful combination therapy employing two individual drugs in vitro and in vivo has encouraged the ongoing development of dual-target inhibitors. Building upon this discovery, Shi et al. designed a novel dual-target inhibitor by combining the G9a inhibitor BIX-01294 and the EZH2 inhibitor Tazematostat, effectively inhibiting the activity of both G9a and EZH2 [123]. It was demonstrated that in an in vitro experiment, Compound 10 exhibited anti-tumor growth effects in RD and SW982 cells. In addition, Compound 11 demonstrated favorable pharmacokinetic properties in the in vivo experiment. However, there is currently a lack of data to elucidate the efficacy of this dual-target inhibitor in prostate cancer. In this regard, we anticipate forthcoming research findings and a more in-depth exploration of the mechanisms to comprehensively understand the role of G9a/EZH2 dual inhibitors in prostate cancer treatment.

### 7.3. Dual Inhibitor of G9a and HDAC

It is observed that G9a interacts with HDAC, thereby contributing to the aberrant global methylation status in cancers [47,49]. Mechanistically, H3K9 acetylation might impede the formation of H3K9 methylation. Hence, the sequential process involving HDAC-mediated removal of acetylation on H3K9, followed by G9a-mediated methylation modification, could dramatically lead to gene silencing. Numerous studies have revealed that the simultaneous inhibition of G9a and HDAC effectively repressed tumor growth and metastasis. This not only highlights the potentiality of targeting them as therapeutic regimens but also emphasizes the need for the continued development of dual-target inhibitors against G9a and HDAC.

Overall, the development of dual-target inhibitors, especially in epigenetic inhibitors for cancer treatment, remains a large blur to explore. Great efforts have been made in investigating the interactions between G9a and EZH2, as well as G9a and HDAC, demonstrating the potential effectiveness of these combination inhibition strategies in anti-tumor therapy (Figure 4). Nonetheless, several challenges persist, particularly in specific cancer types such as prostate cancer, where the actual efficacy of these inhibitors needs further validation.

## 8. Conclusions

Taken together, this review has systematically explored the multifaceted roles of G9a in the pathogenesis and progression of cancer, especially PCa. Through detailed examination, we have delineated the significant contributions of G9a to epigenetic dysregulation in cancer, including the silencing of tumor suppressor genes, modulating critical pathways relevant to tumor development, promoting the cancer lineage plasticity, and hypoxia adaption. Our review focused more attention on the field of PCa. The incidence of research investigating the function of G9a in PCa has greatly increased over the past decade, owing to the development of more potent epigenetic inhibitors for the treatment of prostate cancer. However, current research on G9a in prostate cancer reveals significant gaps in our understanding, particularly regarding its interactions in the TME and its full epigenetic impact. Exploring G9a inhibitors, especially in combination with immunotherapy, offers promising avenues for PCa treatment research. In the future, more integrated multidisciplinary approaches encompassing molecular biology, pharmacology, and clinical oncology will be pivotal in uncovering the full therapeutic potential of targeting G9a in PCa. We hope that this review serves as a valuable source for researchers and clinicians, guiding future investigations in the field of PCa or pan cancers and paving the way for knowledge into more effective treatment strategies.

## Figures and Tables

**Figure 1 cancers-16-02175-f001:**
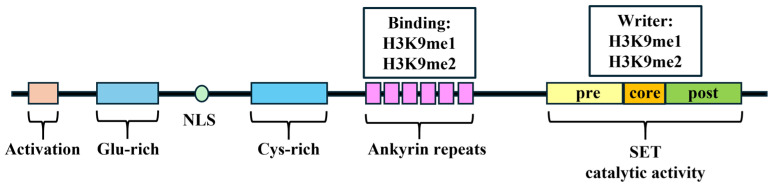
The schematic domains of G9a. Glu: glutamate; NLS: nuclear localization signals; Cys: cysteine. H3K9: histone 3 lysine 9.

**Figure 2 cancers-16-02175-f002:**
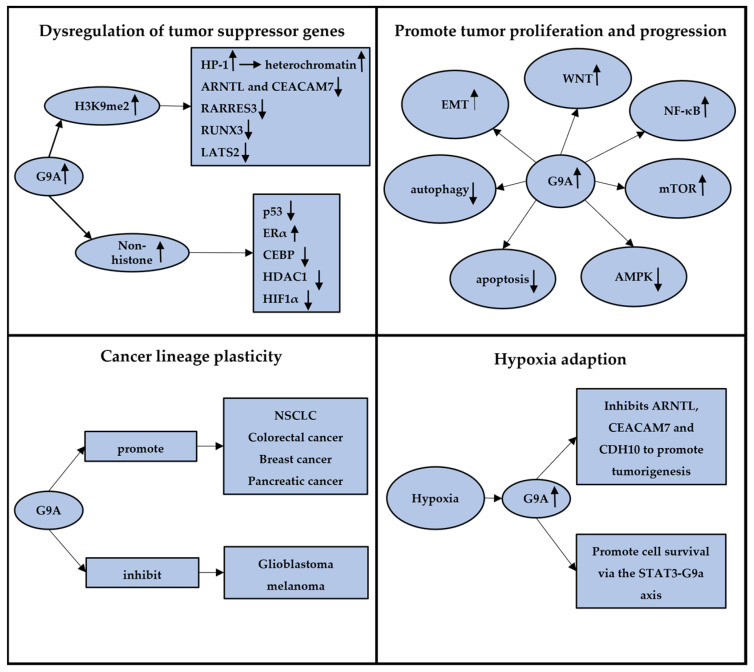
The graphic abstract summarizing four different aspects of G9a in cancers. A, G9a can methylate both histone and non-histone substrates, dysregulating the expression of tumor suppressor genes; B, G9a could impact various pathways in tumors, promoting tumor proliferation and progression; C, G9a plays dual roles in cancer lineage plasticity across various conditions and tumor types; D, G9a plays a crucial role in tumor adaptation to hypoxia, promoting tumor formation and progression. The arrows”↑” and “↓” mean the upregulation and downregulation of genes or pathways.

**Figure 3 cancers-16-02175-f003:**
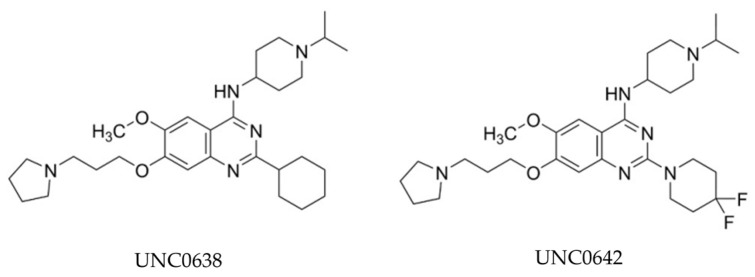
Chemical structures of UNC0638 and UNC0642.

**Figure 4 cancers-16-02175-f004:**
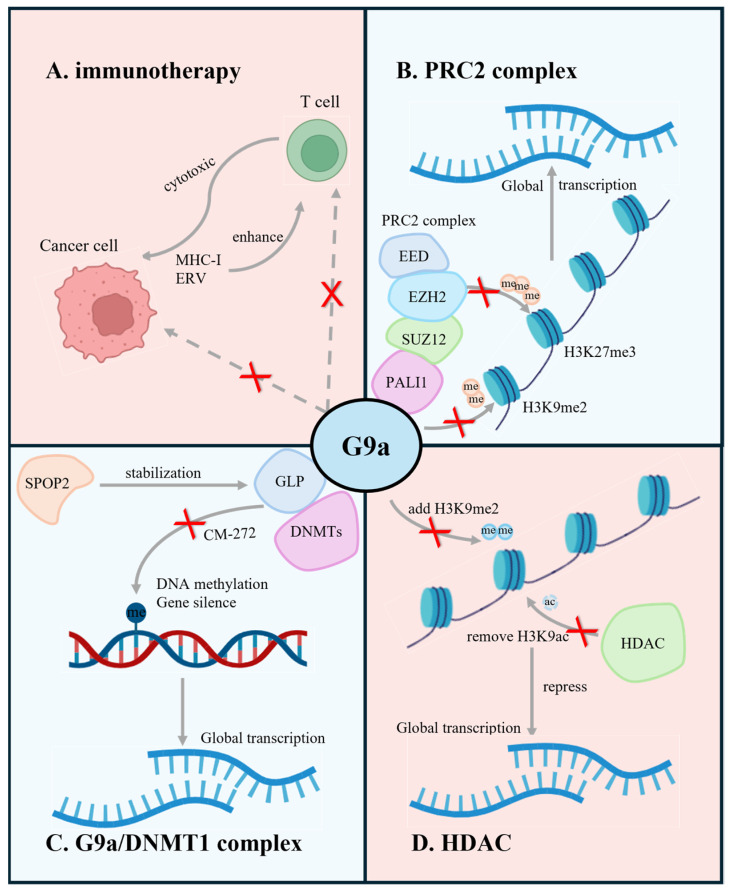
Graphic abstract of the current research (blue, top left and bottom right) and prospective (red, top right and bottom left) of G9a in prostate cancer. Red “x” indicates the inhibition of the respective gene function through different methods. The illustration demonstrates that (**A**) G9a plays a crucial role in tumor immune microenvironment regulation; (**B**) G9a, cooperating with PALI1, SUZ12, EZH2, EED, forms the PRC2 complex to carry out the epigenetic modification in PCa; (**C**) G9a/DNMT1 complex maintains the epigenetic features of PCa through DNA methylation; (**D**) G9a interacting with HDAC contributes to the aberrant global methylation status in cancers. (Created with BioRender.com accessed on 15 May 2024).

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
