# Peer review of "G9a in Cancer: Mechanisms, Therapeutic Advancements, and Clinical Implications"

_cancers, 2024, doi:10.3390/cancers16122175_

Round 1

Reviewer 1 Report

Comments and Suggestions for Authors

            The manuscript entitled “G9a in Cancer: Mechanisms, Therapeutic Advancements, and Clinical Implications” by Ni et al. reviews the role of an important histone methyltransferase in cancer development and describes as well the pharmacological aspects of its inhibitors. The subject has been extensively reviewed, so it is difficult to add novel approaches in a review article. The main strength of the manuscript is the reference to the tumour microenvironment, especially in section 7.1, a question whose interest has been often disregarded. But there are several weaknesses, among which the main one is the absence of some recent data on G9a inhibitors, as detailed below.

            As commented above, many reviews have recently appeared and many of them are not mentioned in the text. As an example the following articles may be mentioned: Nachiyappan et al. EHMT1/EHMT2 in EMT, cancer stemness and drug resistance: emerging evidence and mechanisms. FEBS J. (2022) 289:1329-1351; Haebe et al. Emerging role of G9a in cancer stemness and promises as a therapeutic target. Oncogenesis (2021) 10:76; Souza et al. EHMT2/G9a as an Epigenetic Target in Pediatric and Adult Brain Tumors. Int. J. Mol. Sci. (2021) 22:11292. The authors must, at least, consult these and other similar articles to avoid including in their text too many issues recently reviewed.

            In the specific case of G9a inhibitors, some relevant reviews have been recently published. To give only three examples, the following papers may be mentioned: Li et al. Small molecules targeting selected histone methyltransferases (HMTs) for cancer treatment: Current progress and novel strategies. Eur. J. Med. Chem. (2024) 264:115982; Barghout et al. Chemical biology and pharmacology of histone lysine methylation inhibitors. Biochim. Biophys. Acta Gene Regul. Mech. (2022) 1865:194840; Dhuguru & Ghoneim. Quinazoline Based HDAC Dual Inhibitors as Potential Anti-Cancer Agents. Molecules (2022) 27:2294. The authors must deal with these reviews in the manner mentioned above. Moreover, there are two recently described inhibitors (Zhang et al. Discovery of novel G9a/GLP covalent inhibitors for the treatment of triple-negative breast cancer. Eur. J. Med. Chem. (2023) 261:115841; Jan et al. Discovery of SDS-347 as a specific peptide competitive inhibitor of G9a with promising anti-cancer potential. Biochim. Biophys. Acta Gen. Subj. (2023) 1867:130399) that must be included in the relation given by the authors.

            In summary, the manuscript must be re-written to include the novel relevant data and to abridge the questions mentioned in the most recent reviews.

            Apart from this main objection, several other points need to be addressed:

1) The mention of the work of Pangeni et al. on lines 230 ff. has two errors; first, the genes mentioned are those selected by the authors to validate the genome-wide data but they are mentioned as if they were the only upregulated genes; most importantly, these genes are not upregulated by G9a, but by knocking down G9a.

2) The article of Casciello et al. (ref. 62) is adequate in the context mentioned on lines 286 ff., but not to introduce a general view of hypoxia, as in the first paragraph of section 4.4. The reference No. 30 is improperly quoted on line 283 as Francesco et al. Francesco is the first name of Casciello.

3) A figure summarizing the data given in section 4 may add to follow the wealth of information provided in it.

4) The legend of the graphical abstract of figure 2 needs further explanation. It is difficult to distinguish the “Blue” and “Red” items. Moreover, do the red Xs mean potential sites for inhibitors?

Comments on the Quality of English Language

The text is easily understandable, but moderate editing is required.

Author Response

Dear Reviewer 1:

Thank you for your email and for the constructive comments regarding our manuscript (ID: cancers-2997709) titled "G9a in Cancer: Mechanisms, Therapeutic Advancements, and Clinical Implications." We greatly appreciate the valuable feedback, which has significantly contributed to improving the quality of our review article. We have carefully considered each suggestion and made the necessary revisions accordingly. Please see our point-by-point response to yours comments below. The revised manuscript has been uploaded to the provided link.

We are confident that the revisions have strengthened the manuscript, and we look forward to your positive response.

Comment 1: “many reviews have recently appeared … The authors must, at least, consult these and other similar articles to avoid including in their text too many issues recently reviewed.”

Response: Thank you for pointing this out. According to your suggestion, we have added the following content to the relevant section of the review (lines 51 to 54) as follows:

“Despite recent reviews on G9a primarily focusing on its structure and function, as well as its relationship with stemness, drug resistance, and roles in pediatric and adult brain tumors, significant progress has also been made in studying G9a in the context of prostate cancer and the tumor microenvironment. Herein, in this review…”

Comment 2: “In the specific case of G9a inhibitors, some relevant reviews have been recently published. … that must be included in the relation given by the authors.”

Response: Thanks for your advice. We have added the relevant content and citations in the review as follows:

“Moreover, several new G9a inhibitors have been developed in recent years, building on existing chemical structure cores or featuring entirely new structures, such as ZZM-1220 and SDS-347.”

In addition, we have added the reference you mentioned at the beginning of this section, and added the relevant description of ZZM-1220 in the table 1.

Comment 3: The mention of the work of Pangeni et al. on lines 230 ff. has two errors; first, the genes mentioned are those selected by the authors to validate the genome-wide data but they are mentioned as if they were the only upregulated genes; most importantly, these genes are not upregulated by G9a, but by knocking down G9a.

Response: We apologize for the oversight. As you suggested, we have revised the relevant content in the review as follows:

“Pangeni et al. reported that knocking down of G9a upregulated a series of genes (CDYL2, DPP4, SP5, FOXP1, STAMBPL1, and ROBO1) through hypomethylated H3K9, consequently sustaining tumor-initiating cells and facilitating cancer metastasis.”

Comment 4: The article of Casciello et al. (ref. 62) is adequate in the context mentioned on lines 286 ff., but not to introduce a general view of hypoxia, as in the first paragraph of section 4.4. The reference No. 30 is improperly quoted on line 283 as Francesco et al. Francesco is the first name of Casciello.

Response: We have revised the relevant content in the review as follows:

“In breast cancer, Casciello et al. found that G9a was a key player in mediating the hypoxia-related response. Under hypoxic conditions, an increased level of G9a inhibited several genes, including ARNTL, CEACAM7, and more to promote cancer cell survival and tumorigenesis.”

Comment 5: A figure summarizing the data given in section 4 may add to follow the wealth of information provided in it.

Response: Thanks for your inspiring advice. We have summarized all the important information in this Section 4 and added a new Figure 2 to illustrate the content for easily understanding.

Comment 6: The legend of the graphical abstract of figure 2 needs further explanation. It is difficult to distinguish the “Blue” and “Red” items. Moreover, do the red Xs mean potential sites for inhibitors?

Response: To enhance clarity and ease of understanding, we have updated the figure by changing the background colors to deeper blue and red (Figure 4). Additionally, we have added the related legend to the figure as follows:

“Red ’x’ indicates the inhibition of the respective gene function through different methods.”

Reviewer 2 Report

Comments and Suggestions for Authors

The review by Ni et al., nicely summarizes the current knowledge about G9A in general and its involvement in cancer. It is well written and covers the most important aspects about G9A. I expect that the review will be of interest to a wide audience.

I have only some minor suggestions:

1) The phrase "The ankyrin repeat domain of 33 amino acids" is unclear, because it suggests that the entire ankyrin repeat region is only 33 amino acids. However, the 33 amino acids refer to only one repeat. This should be revised.

2) It may be worth to explain better the role of H3K9 methylation for gene repression. The only sentense about this aspect is "since H3K9 methylation primarily contributes to gene silencing".

3) It may be worth to mention that only G9A, but not GLP, is commonly upregulated in cancer. 

4) The association of G9A with the PRC2 complex, via Pali1, has first been described by PMID: 29628311, which should be cited.

5) The figure 2 can be enlarged. 

6) Based on AlphaFold and PMID: 34278292 the "cysteine-rich region" is a globular domain (https://alphafold.ebi.ac.uk/entry/Q96KQ7). In Prosite this domain is called "zink-binding domain" (https://www.ebi.ac.uk/interpro/protein/UniProt/Q96KQ7/). I would recommended to change this in the text and the figure.

7) In the sentense "Structurally, G9a comprises several domains critical for its function: a SET domain responsible for its methyltransferase activity, an activation domain, a glutamate-rich region, a cysteine-rich region, seven ankyrin repeats, and nuclear localization signals on the N-terminal region (Figure 1)." I would change the order of the domains/regions from the N-terminus to the C-terminus, without jumping around. 

Author Response

Dear Reviewer 2:

Thank you for your email and for the constructive comments regarding our manuscript (ID: cancers-2997709) titled "G9a in Cancer: Mechanisms, Therapeutic Advancements, and Clinical Implications." We greatly appreciate the valuable feedback, which has significantly contributed to improving the quality of our review article. We have carefully considered each suggestion and made the necessary revisions accordingly. Please see our point-by-point response to yours comments below. The revised manuscript has been uploaded to the provided link.

We are confident that the revisions have strengthened the manuscript, and we look forward to your positive response.

Comment 1: The phrase "The ankyrin repeat domain of 33 amino acids" is unclear, because it suggests that the entire ankyrin repeat region is only 33 amino acids. However, the 33 amino acids refer to only one repeat. This should be revised.

Response: Thanks for the detailed comment. As suggested, we have revised the relevant content as follows:

“Six contiguous ankyrin repeats, each consisting of 33 amino acids, form a module in G9a that is capable of binding mono- and dimethyl lysine, thereby mediating protein-protein interactions.”

Comment 2: It may be worth to explain better the role of H3K9 methylation for gene repression. The only sentense about this aspect is "since H3K9 methylation primarily contributes to gene silencing".

Response: Thank you for your great suggestion. We have added more detailed description of the role of H3K9 methylation for gene expression in the review article as follows:

“H3K9 methylation is dynamically regulated by various enzymes and is involved in multiple biological processes, including development, differentiation, and response to environmental signals. This modification plays a critical role in regulating gene expression by contributing to the formation of heterochromatin and the repression of specific genomic regions. Since H3K9 methylation primarily contributes to gene silencing, G9a was generally regarded as an epigenetic repressor.”

Comment 3: It may be worth to mention that only G9A, but not GLP, is commonly upregulated in cancer.

Response: According to your advice, we have added the relevant description about GLP in cancer as follows:

“Notably, overexpression of G9a, and not of its related protein GLP, has often been associated with a more aggressive phenotype in cancer.”

Comment 4: The association of G9A with the PRC2 complex, via Pali1, has first been described by PMID: 29628311, which should be cited.

Response: We have added the following content and the relevant reference as follows:

“Moreover, with further investigation into the structure of the PRC2 complex, G9a was found to be able to bind with the PRC2 complex, forming a novel G9A-PALI1-PRC2 structure and conducting dual H3K9 and H3K27 methylation epigenetic modification.”

Comment 5: The figure 2 can be enlarged.

Response: We have updated the figure.

Comment 6: Based on AlphaFold and PMID: 34278292 the "cysteine-rich region" is a globular domain (https://alphafold.ebi.ac.uk/entry/Q96KQ7). In Prosite this domain is called "zink-binding domain" (https://www.ebi.ac.uk/interpro/protein/UniProt/Q96KQ7/). I would recommended to change this in the text and the figure.

Response: Thanks for your suggestion. The term "zinc-binding domain" is derived from the domain's binding function, while "cysteine-rich domain" is named based on its composition. Both names are correct and reasonable. However, after carefully reviewing the G9a literature, including the first published study in the field of G9a research (PMID: 11316813) and other recent ones (PMID: 34278292 and 34685453), we realized "cysteine-rich domain" is more commonly used in this field. Therefore, we prefer to retain the "cysteine-rich domain" in this review.

Comment 7: In the sentence "Structurally, G9a comprises several domains critical for its function: a SET domain responsible for its methyltransferase activity, an activation domain, a glutamate-rich region, a cysteine-rich region, seven ankyrin repeats, and nuclear localization signals on the N-terminal region (Figure 1)." I would change the order of the domains/regions from the N-terminus to the C-terminus, without jumping around

Response: Thank you for your valuable suggestions. We rewrote the content as follows:

“Structurally, G9a comprises several domains critical for its function: an activation domain, a glutamate-rich region, nuclear localization signals, a cysteine-rich region, six ankyrin repeats, and a SET domain responsible for its methyltransferase activity from the N-terminus to the C-terminus (Figure 1).”

Reviewer 3 Report

Comments and Suggestions for Authors

The manuscript highlights the histone methyltransferase G9a as a promising target for cancer therapy, in particular, for advanced prostate cancers. The content of the manuscript is surely of interest for the readers of this journal. However, the manuscript needs to be revised in order to become suitable for publication in this journal:

Section 4: I suggest to provide a (colored) figure and/or table at the end of this section, which summarizes the detailed content of section 4.

Section 5: What structural modifications led to the described in vivo activity by UNC0642 when compared with the other quinazolines without in vivo activity? Maybe the authors can provide a figure showing the chemical structures and structural differences/modifications.

Table 1: The meaning of the table is unclear and needs more explanation in the header or footnote. The meaning ´´Available in vivo´´ is unclear. ´´Selectivity´´ is unclear, I can only see activities against G9a and GLP, and there is no comment on selectivity.

Section 6: The nuclear receptor AR and its splice variants play a crucial role in prostate cancer. How far interacts G9a with AR function and AR-mediated gene transcription regulation, in particular, of the splice variant AR-V7? Is combination of G9a inhibitors with AR inhibitors and androgen deprivation therapy a useful option for prostate cancer therapy?

Comments on the Quality of English Language

n.a.

Author Response

Dear Reviewer 3:

Thank you for your email and for the constructive comments regarding our manuscript (ID: cancers-2997709) titled "G9a in Cancer: Mechanisms, Therapeutic Advancements, and Clinical Implications." We greatly appreciate the valuable feedback, which has significantly contributed to improving the quality of our review article. We have carefully considered each suggestion and made the necessary revisions accordingly. Please see our point-by-point response to yours comments below. The revised manuscript has been uploaded to the provided link.

We are confident that the revisions have strengthened the manuscript, and we look forward to your positive response.

Comment 1: Section 4: I suggest to provide a (colored) figure and/or table at the end of this section, which summarizes the detailed content of section 4.

Response: Thanks for your inspiring advice. We have summarized all the important information in Section 4 and added a new Figure 2 to facilitate easier understanding of the content.

Comment 2: Section 5: What structural modifications led to the described in vivo activity by UNC0642 when compared with the other quinazolines without in vivo activity? Maybe the authors can provide a figure showing the chemical structures and structural differences/modifications.

Response: Compared to UNC0638, UNC0642 alters the side chain structure and introduces fluorine atoms, enhancing the overall metabolic stability and pharmacokinetic properties of the drug, thereby exhibiting good in-vivo capabilities. We have added a new figure 3 in the review to show the difference between UNC0638 and UNC0642.

Comment 3: Table 1: The meaning of the table is unclear and needs more explanation in the header or footnote. The meaning ´´Available in vivo´´ is unclear. ´´Selectivity´´ is unclear, I can only see activities against G9a and GLP, and there is no comment on selectivity.

Compound

Core

Application in vivo

IC50s

Response: According to your advice, we have carefully revised the header of the Table 1 as follows:

Comment 4: Section 6: The nuclear receptor AR and its splice variants play a crucial role in prostate cancer. How far interacts G9a with AR function and AR-mediated gene transcription regulation, in particular, of the splice variant AR-V7? Is combination of G9a inhibitors with AR inhibitors and androgen deprivation therapy a useful option for prostate cancer therapy?

Response: Thanks for your thoughtful comments. To our knowledge, there is currently no evidence to suggest an interaction between G9a and AR-V7. Regarding to AR, there is only one article from over a decade ago mentioned that G9a could act as a coactivator for AR and reduction of G9a diminished AR activity (PMID: 16451774). Unfortunately, there is no follow-up study regarding the interaction between AR and G9a. Further study will be needed to provide more convincing evidence.

Currently, there is no direct evidence to demonstrate the efficacy of combination therapy of G9a inhibitors and AR inhibitors or androgen deprivation therapy. Interestingly, it has been suggested that H3K9 methylation could drive prostate cancer cells to acquire an ARSI-resistant phenotype (PMID: 35584120). More studies will be needed to clarify their interaction and the potential application of anti-G9a and anti-AR combination therapy. Accordingly, we added the new content below into the review:

“AR inhibitors and androgen deprivation therapy (ADT) are crucial for the clinical treatment of prostate cancer. However, there is currently a lack of solid evidence indicating an interaction between G9a and AR, or even AR-V7. Although one study showed that H3K9 methylation can promote resistance to AR inhibitors in prostate cancer cells and that de-methylation may potentially restore drug sensitivity, it remains unclear whether there is a direct interaction between G9a and AR.”

Round 2

Reviewer 1 Report

Comments and Suggestions for Authors

            The revised version of the manuscript cancers-2997709 by Ni et al. incorporates several of the suggestions made on the former version. Nevertheless, the authors have dealt with the two main concerns I posed in my first report only in a superficial manner. It is true that I mentioned several recent reviews on the field of the manuscript, but I stated: “The authors must, at least, consult these and other similar articles to avoid including in their text too many issues recently reviewed”. In other words, it was not intended that the authors must include these references, a question that they have done, but that they must consult them to avoid reiterations of subjects already reviewed by other authors. These reiterations still persist, so I insist in recommending an abridgement of the questions mentioned in those recent reviews.

            The minor points mentioned in my first report have been satisfactorily addressed, although there are some concerns about Figure 2. Apart from the typo in its legend (summaried instead of summarized), interpretation of the figure may lead to confusion. For instance, dimethylation of ERα, as mentioned in the article of Zhang et al. quoted by the authors “is recognized by the Tudor domain of PHF20, which recruits the MOF histone acetyltransferase (HAT) complex to ERα target gene promoters to deposit histone H4K16 acetylation promoting active transcription”. Taking these data on account, the arrow pointing downwards after ERα may induce to confusion. This is a single example; the authors are urged to check the clarity of the symbols included in the remaining nonhistone proteins modified by G9a.

Comments on the Quality of English Language

Moderate editing is required to avoid typos and grammar errors.

Reviewer 3 Report

Comments and Suggestions for Authors

The revised manuscript is suitable for publication now.

Comments on the Quality of English Language

n.a.

Author Response

Thanks for your great assistance to our review!